# Scouting for Potential LLMs: A Preliminary Assessment of Domain Adaptability for Supervised Fine-Tuning

## Abstract

Large Language Models (LLMs) have demonstrated remarkable performance across diverse tasks, but their effectiveness in domain-specific applications depends on how well the Supervised Fine-Tuning (SFT) data aligns with the model's pre-trained knowledge. Since SFT doesn't always improve performance, developers must resort to costly trial-and-error to find optimal model-dataset matches. To address this problem, we introduce **Potential Scout**, a lightweight framework that diagnoses a model's suitability for SFT without training. Our method builds a **Thinking Curve Matrix (TCM)** that tracks how hidden representations evolve across transformer layers when processing SFT samples. From TCM, we derive two diagnostic indicators: **Activation Growth Score**, which captures how well the model distinguishes semantic differences, and **Layer Coverage Score**, which measures representational stability within the model. Combined with these indicators and pre-SFT benchmark scores, we designed two complementary scouting modes: **In-dataset Scout** uses prior SFT experience on the same dataset, while **Cross-dataset Scout** works on entirely new datasets. Across 18 LLMs and 8 datasets, Potential Scout identifies top-performing candidates in minutes, substantially reducing the search space for SFT and eliminating extensive exploratory experiments in model selection.

## 1 Introduction

Large language models (LLMs) have expanded rapidly beyond language generation, enabling applications across numerous fields. To complete these tasks, LLMs need several capabilities, such as reasoning and domain-specific knowledge (Besta et al., 2025; Zhong et al., 2024; Zhao et al., 2024a). One strategy to grant these capabilities to small-scale LLMs is supervised fine-tuning (SFT) using task-specific data (Liu et al., 2024a; Guo et al., 2025). Recent studies show that smaller models can acquire these capabilities by fine-tuning with high-quality data (Huang et al., 2024; Min et al., 2024; Ye et al., 2025).

However, fine-tuning does not always guarantee benchmark score improvements, even when using relevant data for SFT. Excellent models sometimes perform worse after SFT, while weaker models can achieve substantial gains. This phenomenon is illustrated across different benchmarks in Figure 1. On the MATH500 benchmark, Qwen2.5-Math-1.5B shows a slight improvement after SFT with the LIMO dataset, while the larger Qwen2.5-3B suffers a performance drop, although its scale is larger than the math model (Ye et al., 2025; Yang et al., 2024a;b). Similarly, on GSM8K, although DeepSeek-Math-7B-Instruct showed the weakest pre-SFT performance, it achieved the largest relative improvement (+235%) and ultimately recorded the highest post-SFT accuracy (Cobbe et al., 2021; Zhihong Shao, 2024). These results suggest that compatibility between prior knowledge of the model and the SFT data is a key factor in fine-tuning. However, users typically lack access to pretraining corpora, making it difficult to assess this alignment (Yan et al., 2024). This often forces users to resort to costly trial-and-error approaches, which require exhaustive SFT training and evaluation across multiple candidate models to identify the optimal LLM.

To address this issue, we introduce **Potential Scout**, a scouting framework that assesses the potential for fine-tuning by analyzing how hidden states evolve across transformer layers in response to SFT

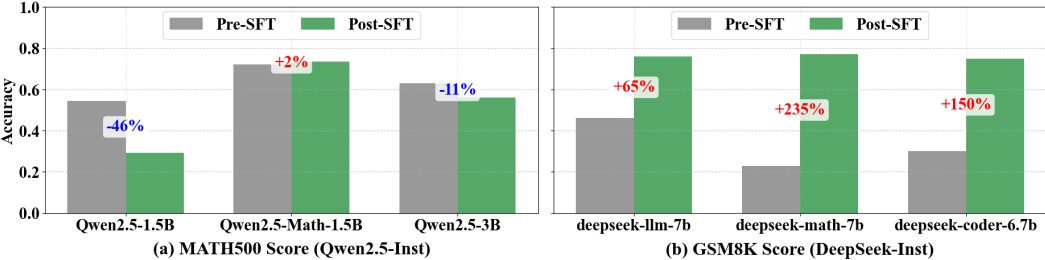

Figure 1: Evaluation before and after SFT on two benchmarks, showing that alignment between a model's prior knowledge and the SFT data is a key factor in fine-tuning success. (a) MATH500 results after SFT with the LIMO (math reasoning) dataset. Although Qwen2.5-Math-1.5B shows a slight improvement, the larger Qwen2.5-3B suffers a performance drop. (b) GSM8K results after SFT with the GSM8K training set. Notably, DeepSeek-Math-7B, despite having the weakest pre-SFT performance, achieves the largest relative improvement (+235%) and the highest post-SFT accuracy, while models with stronger initial performance show more modest gains.

data samples through a Thinking Curve (TC). The TC represents a trajectory of change in semantic representation for a single query as it passes through consecutive model layers. To systematically explore these patterns, we construct a **Thinking Curve Matrix (TCM)** that captures several queries from the same dataset and records their hidden states layer by layer.

From the TCM, we compute two scouting indicators: **Activation Growth Score (AGS)** and **Layer Coverage Score (LCS)**. *AGS* indicates the power of semantic expansion through layers, based on the findings that fine-tuned models show a stronger layer-wise feature specialization (Nadipalli, 2025; Zhao et al., 2024b). *LCS* quantifies a stability of semantic differentiation across queries at each layer, based on studies showing that domain-specialized models exhibit more structured representational patterns (Zhou & Srikumar, 2021; Phang et al., 2021).

Combined with pre-SFT benchmark scores, these indicators enable post-SFT performance prediction through two **scouting modes**: **(1) In-dataset Scout**, which uses past SFT experience on the same dataset to assess candidate models, and **(2) Cross-dataset Scout**, which works on entirely new datasets where no SFT results exist. Through evaluation across 18 LLMs and 8 datasets, we show that *Potential Scout* correctly identifies about 70% of top-performing models, reducing selection time from days of training to minutes of analysis. By transforming model selection into a systematic diagnostic process, Potential Scout eliminates extensive exploratory experiments and **accelerates the development of specialized LLMs across diverse domains**. The contributions of this work are summarized as follows:

- **Preliminary Assessment for Model Selection**: We propose a scouting framework that enables the identification of promising models before SFT, reducing costly and extensive training and testing by analyzing internal representation.
- **Sample-Efficient Evaluation**: Our method requires only a small fraction of dataset samples (e.g., 5%, corresponding to 15 to 350 instances), allowing reliable prediction of the fine-tuning potential with minimal computational overhead (7 to 8 minutes per model).
- **Dual Scouting Approaches**: In-dataset Scout utilizes dataset-specific experience to achieve 70% Top-7 precision among 18 candidate models, and Cross-dataset Scout enables model selection in entirely new datasets with 70% Top-9 precision.

## 2 RELATED WORK

### 2.1 EFFICIENT SUPERVISED FINE-TUNING

SFT has been widely used to adapt LLMs for specific domains, but its high computational cost has driven extensive research on efficiency. One major approach is knowledge distillation, where smaller student models replicate larger teachers through output distribution matching (Hinton et al., 2015; Ba & Caruana, 2014; Romero et al., 2014). Recent work explores SFT distillation, where students

are fine-tuned on teacher-synthesized data (Li & Mooney, 2024; Chen et al., 2024) or self-generated corpora (Wang et al., 2022; Li et al., 2023). A second approach focuses on parameter-efficient fine-tuning (PEFT), which adapts LLMs by training only a fraction of parameters while freezing most weights. Hu et al. (2022) introduce low-rank updates achieving near full fine-tuning performance with minimal overhead, Dettmers et al. (2023) combines 4-bit quantization with LoRA for memory efficiency, and subsequent work explores sub-4-bit quantization and adapter architectures like LLM-Adapters (Kim et al., 2023; Hu et al., 2023). Data selection methods form a third field of research, improving efficiency by curating informative training samples. Liu et al. (2024b) distills domain knowledge through sample re-weighting, Yang et al. (2024c) uses small proxy model trajectories for selection guidance, Chen et al. (2025) maximizes information gain in semantic space for instruction tuning, and Pan et al. (2024) employs gradient-based criteria for quality and diversity. While these approaches significantly reduce SFT costs, they assume the target model is already chosen. Our work addresses a complementary problem: **how to identify the best model candidates for SFT using a training-free method.**

### 2.2 Analyzing Hidden Representations in LLMs

While prior studies have focused on modifying models through pruning or editing, we analyze hidden activations to assess dataset-specific information organization and rank candidate models by their fine-tuning potential. As LLMs have grown in size and complexity, understanding their internal behavior has become essential to improve interpretability, efficiency, and transferability. Many studies have explored hidden representations to make LLMs efficient by uncovering some redundancy from each layer (Michel et al., 2019; Liang et al., 2025; Pons et al., 2024; Lu et al., 2024). Similarity-based approaches, such as CKA, CCA, and SVCCA (Kornblith et al., 2019; Raghu et al., 2017), have been proposed to analyze layer-wise alignment for pruning or distillation purposes. Some studies have considered whether it is possible to edit knowledge of LLMs by intervening in specific internal components, rather than retraining the entire model such as ROME (Meng et al., 2022) and PMET (Li et al., 2024). Additionally, recent work has shown that LLMs can exhibit distinct hidden state patterns depending on their specialized domain. Garcia et al. (2025) demonstrate that inputs from different domains (e.g., mathematics, law, medicine) produce consistent and separable activation trajectories across layers.

## 3 Methodology

Our framework operates through a three-stage process, as demonstrated in Figure 2. First, we extract two diagnostic indicators, **Activation Growth Score** and **Layer Coverage Score**, from candidate models using SFT dataset samples, combined with baseline performance from pre-SFT benchmarks (Figure 2-a). Second, we optimize our dual scouting system using these features: In-dataset Scout for familiar datasets and Cross-dataset Scout for new domains (Figure 2-b). Finally, both scouts generate potential rankings that predict fine-tuning success, as shown in the MBPP and GSM8K prediction results (Figure 2-c).

### 3.1 Thinking Curve Matrix

To quantify how models process dataset-specific information, we design the Thinking Curve Matrix (TCM), which systematically captures layer-wise representational changes across multiple queries. This matrix serves as the foundation for extracting our predictive features by revealing both individual query processing patterns and cross-query consistency within a dataset. The TCM construction begins with measuring dispersion scores that quantify the spread of hidden state representations, then tracking layer-wise trajectories, and finally organizing these measurements into a structured matrix format.

Giraldo et al. (2014) introduced a matrix-based entropy measure that quantifies the dispersion of high-dimensional vectors by analyzing eigenvalue distributions. Motivated by this approach, we measure matrix compression levels across transformer layers using eigenvalue-based analysis. Skean et al. (2024) reported that transformers exhibit a V-shaped trajectory, first compressing inputs, then re-expanding for finer semantics. For token embeddings $H \in \mathbb{R}^{T \times D}$, we compute the Gram matrix $G = HH^\top$, extract eigenvalues $\{\lambda_i\}_{i=1}^T$, and normalize them into proportions $p_i$. To

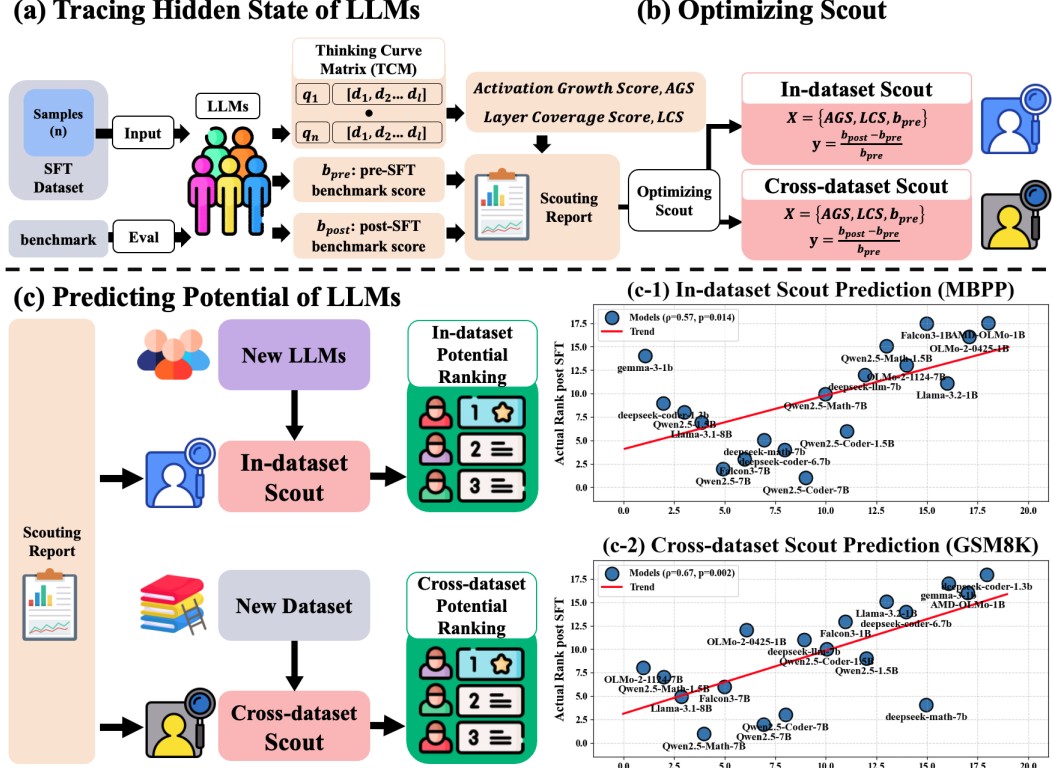

Figure 2: Overview of Potential Scout for identifying promising SFT models. **(a)** Thinking Curve Matrices (TCM) capture hidden-state trajectories to derive *AGS* and *LCS* along with benchmark scores. **(b)** These indicators, stored in a Scouting Report, are used to optimize scouts for predicting performance changes. **(c)** IDS applies when sufficient SFT experience on a dataset exists in Scouting Report, while CDS utilizes other datasets when such information is absent. **(c-1, c-2)** Predicted performance strongly correlates with actual post-SFT performance.

measure the dispersion score, which quantifies the level of semantic differentiation at each layer, the dispersion score is:

$$d(H) = \frac{1 - \sum_{i=1}^{T} p_i^2}{1 - \frac{1}{T} + \varepsilon}. \tag{1}$$

The numerator captures the dispersion by converting concentration measures, while the denominator normalizes across different sequence lengths. Higher values indicate a greater representational spread, with $d(H) \in [0, 1)$. Then, we define the **Thinking Curve (TC)** as the dispersion trajectory across all layers: $TC = \big( d(H^{(1)}),\ d(H^{(2)}),\ \ldots,\ d(H^{(L)}) \big)$, where $H^{(\ell)}$ represents the hidden state at layer $\ell$. As models have different depths, we interpolate all curves to a uniform length of $K = 25$, maintaining the original patterns with $R^2 > 0.95$ quality. We then collect TC from multiple inputs to create the **Thinking Curve Matrix (TCM)**. This matrix reveals both how individual inputs are processed and how consistently the model handles similar queries.

### 3.2 ACTIVATION GROWTH SCORE

Individual TC from the TCM on the GSM8K dataset are visualized as shown in Figure 3. The figure shows TC curves from 30 random samples per model, illustrating their compression–expansion patterns. We measure the **Activation Growth Score (AGS)** of the semantic expansion by computing the slope of the thinking curves after compression. For each sample, we identify the layer of maximum compression (marked with dots in Figure 3) and compute the second half slope of the expansion range, where $x$ represents the dispersion score at each layer. The *AGS* value is then averaged

across all $N$ samples in the dataset:

$$\text{AGS} = \frac{1}{N} \sum_{i=1}^{N} \frac{x_i^{(\ell)} - x_i^{(k)}}{\ell - k}, \qquad k = \text{middle layer of the expansion segment.} \qquad (2)$$

where $x_i^{(\ell)}$ and $x_i^{(k)}$ are the dispersion scores for sample $i$ at layers $\ell$ and $k$ respectively. We use the second half of this expansion segment because previous work shows that dataset-specific expansion emerges mainly in later layers (Garcia et al., 2025; Geva et al., 2020), and this segment also exhibits more pronounced differences between individual samples, making it more informative for distinguishing model capabilities. As shown in Figure 3, we compute AGS only from layers to the right of the vertical blue boundary marking the midpoint between the compression minimum and the final layer. Fine-tuning increases representational differentiation in higher layers (Zhao et al., 2024b), so

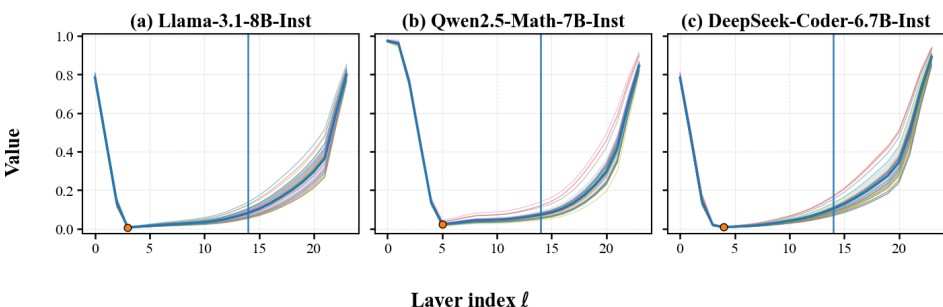

Figure 3: Thinking Curves (TC) of various models on the GSM8K dataset. For each model, we plot curves from 30 randomly selected samples, showing how internal representations evolve across layers. Markers indicate the layer index where each curve reaches its minimum value.

higher AGS indicates stronger expansion capacity, while weaker slopes represent underdeveloped representational structures that are more likely to benefit from SFT.

### 3.3 LAYER COVERAGE SCORE

We measure the **Layer Coverage Score (LCS)** of semantic expansion by computing the coefficient of variation (CV) across columns of the TCM. This captures how consistently a model processes different queries from the same dataset at each layer. Following the same segmentation approach as *AGS* computation, we focus on the second half of the tail region to capture the most informative expansion phase. For each layer $i$ in the selected segment from layer $k$ to $\ell$, we compute the coefficient of variation across all $N$ queries, where $d_j^{(i)}$ represents the dispersion score of query $j$ at layer $i$. The final *LCS* is the average CV across all layers in the segment:

$$\text{LCS} = \frac{1}{\ell - k} \sum_{i=k}^{\ell} \frac{\sigma(d_1^{(i)}, d_2^{(i)}, \ldots, d_N^{(i)})}{\mu(d_1^{(i)}, d_2^{(i)}, \ldots, d_N^{(i)}) + \varepsilon} \qquad (3)$$

This approach is motivated by recent findings that fine-tuning restructures and organizes knowledge by increasing **cohesion within similar knowledge clusters** and improving separation between processing pathways (Zhou & Srikumar, 2021; Doimo et al., 2024). This structural reorganization underlies performance improvements (Zhao et al., 2024b). Models that process similar queries inconsistently have greater potential to improve during fine-tuning. As shown in Figure 4, layers with widely spread values indicate such inconsistency across queries. The same segmentation boundary (indicated by the vertical blue line in Figure 3) is used for LCS, and all computations are performed only on layers to the right of this boundary.

A lower *LCS* indicates that the model processes queries more uniformly across the dataset. However, models with higher variability in their representational patterns actually demonstrate greater fine-tuning potential, as this inconsistency provides substantial room for the organizational restructuring that drives fine-tuning improvements. Combined with *AGS*, these two indicators allow us to comprehensively evaluate a model's fine-tuning potential for specific datasets.

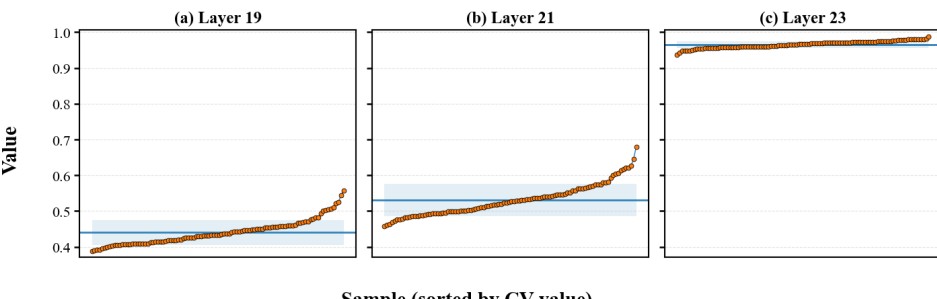

Figure 4: Coefficient of Variation (CV) across layers 19, 21, and 23 for Falcon3-7B-Inst model on GSM8K. Orange dots represent individual query processing, while shaded areas show variance around the mean. Higher spread indicates inconsistent processing of similar queries.

### 3.4 POTENTIAL RANKING VIA LINEAR REGRESSION

**In-dataset Scout (IDS)**    We fit an ordinary least squares regression model to predict the relative improvement after SFT from *AGS* ($g$), *LCS* ($c$), and pre-SFT performance ($b_{\text{pre}}$). For each model $i$ and dataset $j$, the improvement is modeled as

$$\Delta_{ij} = \frac{\text{post}_{ij} - \text{pre}_{ij}}{\text{pre}_{ij}} = \beta_0^{(j)} + \beta_g^{(j)} g_{ij} + \beta_c^{(j)} c_{ij} + \beta_b^{(j)} b_{\text{pre},ij} + \epsilon. \tag{4}$$

The coefficients $\beta^{(j)}$ are estimated separately for each dataset $j$. IDS therefore learns dataset-specific regressions that relate our indicators to fine-tuning gains and becomes stable once roughly ten model observations are available. However, because IDS must be trained separately for each dataset, it cannot be applied to a completely new dataset with no SFT history.

**Cross-dataset Scout (CDS)**    The IDS approach has a critical limitation: it requires separate training for each dataset, making it impractical for new datasets with no fine-tuning history. To address this cold-start problem and enable cross-dataset generalization, we extend IDS using a Linear Mixed Model as a **Cross-dataset Scout (CDS)** that leverages information from multiple datasets.

$$\Delta_{ij} = \beta_0 + \beta_g g_{ij} + \beta_c c_{ij} + \beta_b b_{\text{pre},ij} + \gamma_j + \epsilon, \qquad \gamma_j \sim \mathcal{N}(0, \sigma_\gamma^2). \tag{5}$$

$$\gamma_{\text{unseen}} = 0 \quad \text{for unseen datasets.}$$

The variables $\Delta$, $g$, $c$, and $b_{\text{pre}}$ denote the improvement, AGS, LCS, and pre-SFT performance for each model–dataset pair. The term $\gamma_j$ is a dataset-specific random intercept, modeled as $\mathcal{N}(0, \sigma_\gamma^2)$, which captures systematic differences across datasets. The fixed effects ($\beta_g, \beta_c, \beta_b$) describe relationships between our indicators and fine-tuning gains that generalize across datasets. For unseen datasets, we make predictions using only the fixed-effects component by setting $\gamma_{\text{unseen}} = 0$, allowing CDS to transfer knowledge from previous datasets and operate effectively in cold-start scenarios.

**Overall Prediction Pipeline**    Our pipeline follows a simple four-step process. (1) Extract AGS, LCS, and pre-SFT performance for each candidate model. (2) Use IDS or CDS to predict the model's expected relative improvement. (3) Estimate its post-SFT performance by applying the predicted gain to the pre-SFT score. (4) Rank all models based on these estimated post-SFT performances. This unified pipeline is used for both IDS and CDS.

## 4 EXPERIMENTS

### 4.1 SETUP

All experiments are conducted on two NVIDIA A100 GPUs (80GB each). We evaluated **18 open-source LLMs** using two scales: 9 small-scale models (1–1.5B) and 9 mid-scale models (7–8B), including both general-purpose and domain-specialized instruction-tuned models. The small group includes Qwen2.5 (general, math, coder) (Yang et al., 2024a;b; Hui et al., 2024),

LLaMA-3.2-1B (Grattafiori et al., 2024; Meta AI, 2024), Falcon-3-1B (Team, 2024), AMD-OLMo-1B-SFT (Liu et al., 2024c), OLMo-2-1B (OLMo et al., 2024), Gemma-3-1B (Team et al., 2025), and DeepSeek-Coder-1.3B (Guo et al., 2024). The middle group covers Qwen2.5 (general, math, coder), LLaMA-3.1-8B, Falcon-3-7B, OLMo-2-7B, and DeepSeek (general, math, coder) (Bi et al., 2024; Shao et al., 2024). With these models, we evaluate across three categories of benchmarks: **math** (GSM8K (Cobbe et al., 2021), MATH (Hendrycks et al., 2021; Lewkowycz et al., 2022), MathQA (Amini et al., 2019)), **coding** (MBPP (Austin et al., 2021), LeetCodeDataset (Xia et al., 2025)), and **general QA** (CoQA (Reddy et al., 2019), OpenBookQA (Mihaylov et al., 2018), ARC-Challenge (Clark et al., 2018)).

## 4.2 COMPLEMENTARY MEASUREMENTS AND COMPUTATIONAL EFFICIENCY

**Predictor Complementarity** To assess whether combining all three measurements improves prediction capability, we evaluated their joint performance against individual measurements. The combined regression model consistently achieves the highest correlation with SFT improvement, producing statistically significant relationships $p < 0.05$ in 5 out of 8 datasets with strong correlations: GSM8K $\rho = 0.622$, MathQA $\rho = 0.658$, MBPP $\rho = 0.742$, LeetCode $\rho = 0.471$, and CoQA $\rho = 0.517$, as shown in Table 1. While individual measurements exhibit domain-specific effectiveness, their combination substantially enhances predictive capability, for example, CoQA correlation increases from $0.357$ (best individual predictor) to $0.517$ (combined model). VIF analysis confirms that all measurements maintain scores below 5, indicating minimal multicollinearity (see Appendix B). The regression coefficients broadly align with our expectations. Most $\beta_g$ values are negative, suggesting weaker semantic expansion correlates with higher SFT potential. The $\beta_c$ coefficients are positive in 6 out of 8 datasets, indicating that processing inconsistency generally signals improvement opportunities. Similarly, most $\beta_{b_{pre}}$ values are negative, supporting the intuition that lower baseline performance predicts greater improvement potential.

Table 1: Pearson correlation coefficients (corr), $p$-values (p-val), and IDS coefficients ($\beta$) for individual predictors and their combination. The highest correlation per dataset is in **bold**; the second-highest is underlined. The $\beta$ values are the optimized weights combining all three predictors in IDS. **All** indicates the correlation and $p$-value when using all three predictors jointly.

| Dataset | AGS | | LCS | | $b_{pre}$ | | All | | $\beta$ (All) | | |
|---------|------|-------|------|-------|------|-------|------|-------|-----------|-----------|----------------|
| | corr | $p$-val | corr | $p$-val | corr | $p$-val | corr | $p$-val | $\beta_g$ | $\beta_c$ | $\beta_{b_{pre}}$ |
| GSM8K | 0.048 | 0.851 | 0.316 | 0.202 | 0.516 | 0.028 | **0.622** | 0.006 | -0.53 | +0.57 | -0.45 |
| MATH | 0.170 | 0.499 | 0.328 | 0.184 | 0.202 | 0.421 | **0.417** | 0.085 | -0.11 | +0.44 | -0.24 |
| MathQA | 0.003 | 0.989 | 0.081 | 0.748 | 0.648 | 0.004 | **0.658** | 0.003 | -0.15 | +0.19 | -0.65 |
| MBPP | 0.011 | 0.964 | 0.092 | 0.717 | 0.713 | 0.001 | **0.742** | 0.001 | +0.10 | +0.13 | -0.75 |
| LeetCode | 0.182 | 0.470 | 0.058 | 0.819 | 0.440 | 0.067 | **0.471** | 0.048 | -0.36 | +0.29 | -0.39 |
| CoQA | 0.357 | 0.146 | 0.262 | 0.294 | 0.254 | 0.308 | **0.517** | 0.028 | -0.40 | -0.08 | -0.40 |
| OpenBookQA | 0.254 | 0.309 | 0.196 | 0.435 | 0.025 | 0.923 | **0.279** | 0.262 | -0.23 | -0.12 | -0.14 |
| ARC-Challenge | 0.182 | 0.470 | 0.019 | 0.941 | 0.224 | 0.371 | **0.360** | 0.143 | -0.41 | +0.42 | +0.28 |

**Sampling Efficiency** To balance computational cost with measurement reliability, we evaluated how sampling rates affect our quantitative indicators. Both *AGS* and *LCS* achieve a coefficient of variation (CV) around 0.1 across sampling rates, which means that the indicators remain stable even with limited data. We used datasets that range from 300 to 7,000 samples, where sampling 5% translates into 15 to 350 absolute samples, which proves sufficient for reliable measurement. As summarized in Table 2, using 5% sampling requires only 7-8 minutes per model with a single forward pass for hidden state collection.

Table 2: Computation time per model (minutes). Benchmark refers to a full evaluation, and Potential Scout indicates using a sampled subset and one forward pass to compute *AGS* and *LCS*.

| Method | Sampling | Time(m) |
|--------|----------|---------|
| Benchmark | – | 10 to 180 |
| | 1% | 1 to 3 |
| Potential Scout | 5% | 7 to 8 |
| | 10% | 10 to 15 |

### 4.3 GENERALIZATION ACROSS MODELS AND DATASETS

We evaluated Potential Scout's generalization performance within and across datasets using Top-K precision metrics, as shown in Figure 5. For IDS, we performed K-fold cross-validation within each dataset, splitting models into training and testing folds to predict relative improvement using OLS regression. For CDS, we employed Leave-One-Dataset-Out cross-validation with Mixed-Effects Linear Models, training on all datasets except one, and predicting for the held-out dataset to simulate completely new dataset scenarios. Both approaches used these predictions to estimate post-SFT scores, then ranked models accordingly and measured how well the predicted top-K performers overlapped with actual top-K models.

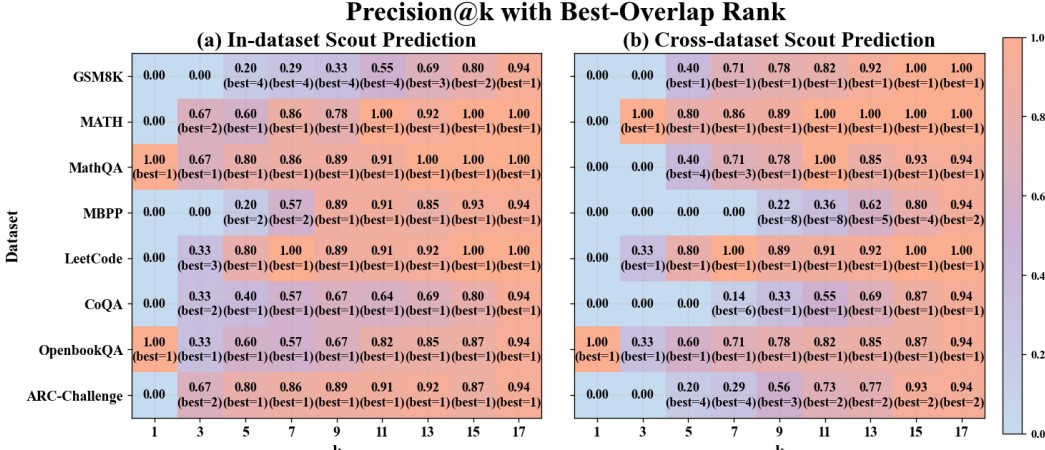

Figure 5: **In-dataset vs. Cross-dataset Scout Performance.** Top-K precision comparison between (a) IDS for datasets with existing SFT experience and (b) CDS for entirely new datasets. While both struggle with Top-1 precision, IDS achieves 70% accuracy at Top-7 and CDS at Top-9 with 18 LLMs. The results demonstrate that effective model scouting remains feasible even without prior SFT experience, though cross-dataset generalization requires consideration of dataset compatibility.

Both approaches struggle with Top-1 prediction, but achieve reasonable performance at moderate k values. Among the 18 candidate models, IDS achieves 70% precision at Top-7, while CDS reaches a similar precision at Top-9. This indicates that while identifying the best model remains challenging, both methods reliably select a small set of high-performing candidates. IDS demonstrates strong performance across most datasets, with precision often reaching 1.0 at higher k values. Exceptions such as GSM8K and MBPP show lower precision at small k but improve substantially as k increases, suggesting that once post-SFT scores are available for several models in a dataset, IDS can accurately identify the remaining top performers with minimal additional evaluation.

CDS shows the more challenging scenario of predicting performance on completely new datasets. Although generally achieving lower precision than IDS with greater variability, it still provides valuable model selection guidance. Performance variation reflects the importance of dataset compatibility—effectiveness depends on how well training datasets align with target dataset characteristics, requiring careful consideration of dataset similarity for cross-dataset generalization.

### 4.4 OPTIMAL k SELECTION AND TRANSITION STRATEGY

CDS enables predictions for completely new datasets by using experience from other datasets. However, once sufficient SFT experience accumulates within the target dataset, building a dataset-specific IDS becomes more beneficial, as IDS provides better performance and stability as demonstrated in Figure 5. This raises two practical questions: when should we transition from CDS to IDS, and what prediction depth (k) reliably captures the best-performing model regardless of the total number of candidates.

To address these questions, we systematically experimented with the performance of IDS in different models' pool sizes and prediction depths. For each dataset, we randomly sampled 6-12 models, used

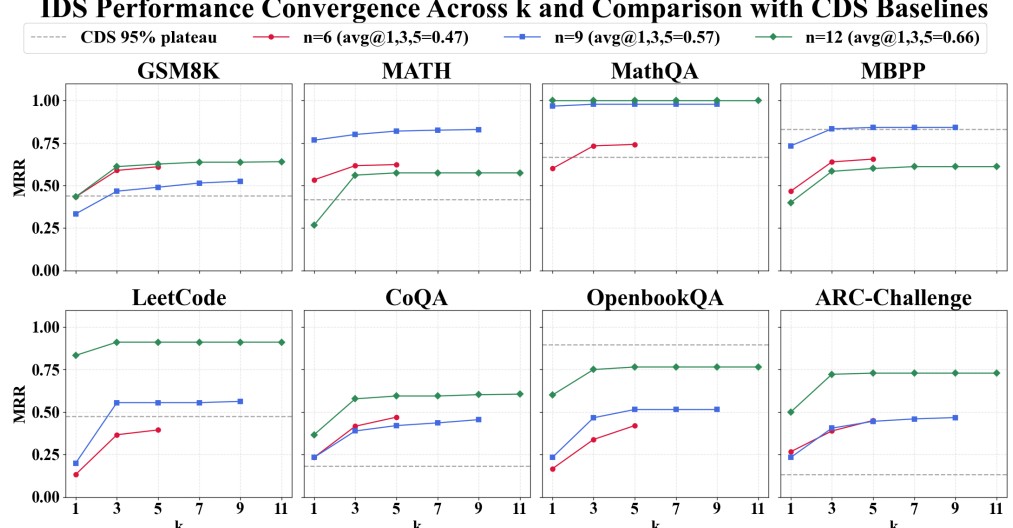

Figure 6: Mean Reciprocal Rank (MRR) performance of IDS across different model pool sizes and prediction depths (k). The horizontal dashed lines represent average CDS performance baselines.

most for training and one for validation, then measured the ranking accuracy using Mean Reciprocal Rank (MRR), where values closer to 1 indicate that the top-performing model is ranked higher. We repeated this process 30 times for statistical reliability.

The results reveal two key patterns in Figure 6. First, MRR consistently converges at k=3 across most datasets, indicating that top-3 predictions are sufficient to identify top-1 within available model pools. Second, IDS performance depends critically on training set size: with only 6 models, IDS struggles to outperform CDS, especially for LeetCode and OpenBookQA. However, with 12 models, IDS exceeds or closely approaches CDS baselines across most datasets.

Our parallel experiments with CDS using different training dataset combinations showed no consistent improvement when adding more datasets, often leading to performance degradation (detailed analysis in Appendix C). This reinforces that dataset compatibility matters more than quantity for cross-dataset knowledge transfer. Based on these findings, we recommend a practical three-stage strategy: (1) use CDS initially for completely new datasets, (2) transition to IDS with k=3 once approximately 10 model observations are available, and (3) for borderline cases (8-10 models), combine both approaches for more robust guidance.

## 5 CONCLUSION

In this work, we propose **Potential Scout**, a framework that fundamentally changes how we select models for fine-tuning by analyzing their internal semantic processing capabilities. Our key insight shows that we can diagnose a model's fine-tuning potential through its **Thinking Curve Matrix (TCM)**, which reveals how effectively the model expands and stabilizes semantic representations. Our two diagnostic indicators capture complementary aspects of model readiness: *Activation Growth Score* reveals whether a model can meaningfully differentiate semantic nuances, while *Layer Coverage Score* shows whether it reliably processes diverse inputs. These indicators provide a principled way to assess whether a model's internal architecture suits domain adaptation. Our dual scouting approach addresses a practical reality: prediction strategies can adapt according to available SFT experience. **In-dataset Scout** utilizes empirical patterns when prior experience exists, while **Cross-dataset Scout** assesses completely new domains by transferring diagnostic insights across datasets. The contribution of this work can transform model selection from extensive exploratory experiments into efficient, data-driven diagnostics. Potential Scout enables developers to identify promising candidates in minutes with much less effort than full days of training, thus reducing the time and cost of selecting optimal models and supporting the efficient development of specialized LLM assistants.

ETHICS STATEMENT

The research carried out in this paper is fully in compliance with the ICLR Code of Ethics.

REPRODUCIBILITY STATEMENT

For our submission, we have uploaded the complete source code as supplementary material. The code includes comprehensive documentation detailing the implementation of the Thinking Curve Matrix (TCM), *Activation Growth Score* and *Layer Coverage Score* indicators calculation, and both IDS and CDS scouting approaches. The codebase contains detailed instructions for dataset preparation, model evaluation, and metric computation across all tested domains and models.

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

APPENDIX

## A  THE USE OF LARGE LANGUAGE MODELS

We used Large Language Models (LLMs) in this research as general-purpose tools for writing enhancement and literature discovery. Specifically, we employed LLMs to: **Writing assistance:** We used LLMs to improve the clarity and fluency of our manuscript, including proofreading, grammar checking, and suggesting alternative phrasings for better readability. **Literature discovery:** LLMs helped us identify relevant research papers and understand connections between different works in the field. However, we independently verified and critically evaluated all cited works before inclusion. We conducted the fundamental research contributions entirely without LLM involvement in the creative or analytical processes. **All fundamental aspects of this research, including the conception of Potential Scout, the design of the Thinking Curve Matrix, the scouting methodology and all analyses, were independently developed by the authors without any contribution from LLMs.**

## B  VARIANCE INFLATION FACTOR (VIF) ANALYSIS

We standardized all predictors (AGS, LCS, and $b_{pre}$) to zero mean and unit variance prior to regression. Then, we computed the variance inflation factor (VIF) for each dataset to verify that the three predictors can be used together in regression. As shown in Figure 7, all variables remain below the conventional threshold of 5, demonstrating that the predictors are sufficiently independent for joint use in regression modeling.

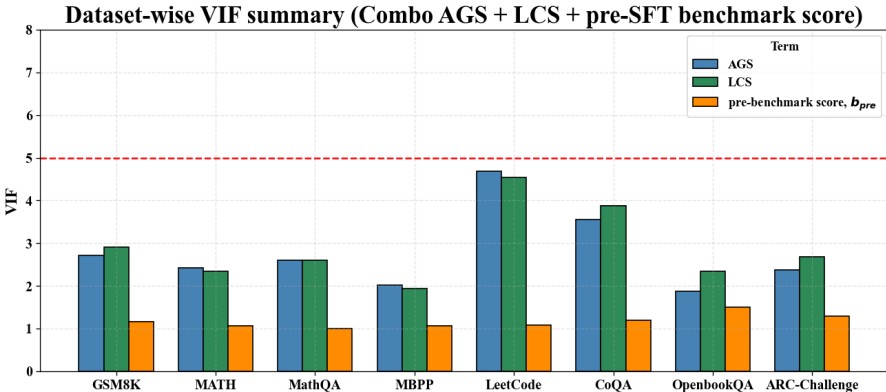

Figure 7: Variance inflation factor (VIF) analysis across datasets for the three predictors (*AGS*, *LCS*, and $b_{pre}$).

## C  ADDITIONAL RESULTS: CDS WITH VARYING TRAINING DATASETS

We further analyzed CDS performance when varying the number of training datasets ($m$). As shown in Figure 8, simply increasing training datasets does not improve performance. In datasets such as **GSM8K**, **MATH**, **LeetCode**, **CoQA**, and **ARC-Challenge**, adding more datasets yields little to no improvement. More critically, for **GSM8K** and **MBPP**, performance actually decreases with additional datasets. These results indicate that CDS works poorly when unrelated datasets are combined. The **quality and compatibility** of datasets matter far more than quantity, confirming that effective cross-dataset supervision requires careful selection of related datasets rather than larger pools of unrelated sources.

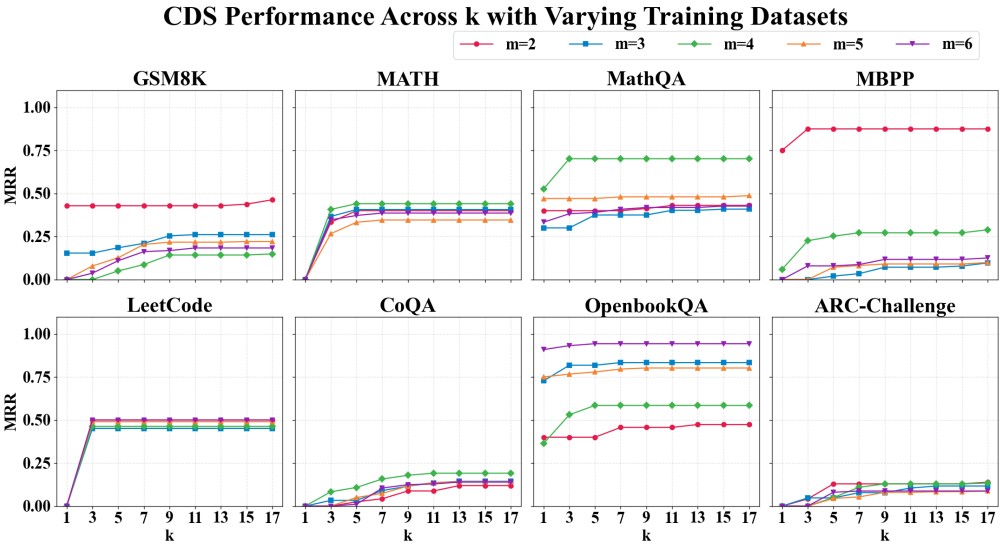

Figure 8: CDS MRR with varying numbers of training datasets ($m$) across different values of $k$.

