# OpenReview forum: "Scouting for Potential LLMs: A Preliminary Assessment of Domain Adaptability for Supervised Fine-Tuning"
_ICLR.cc/2026/Conference — Submitted to ICLR 2026_

### Official Review · Reviewer_6hgC · 2025-10-26

**Soundness:** 4
**Presentation:** 2
**Contribution:** 3
**Rating:** 6
**Confidence:** 4

**Summary:**

This work proposes a training-free method to estimate LLM's potential performance after SFTing on certain datasets. To be specific, they use the eigenvalue extract from the gram matrix of the token embeddings to compute AGS & LCS and further train a small regression model for IDS and CDS, evaluating LLMs' potential from different perspectives. Experiments on 18 LLMs and 8 datasets testify to the effectiveness and efficiency of their method.

**Strengths:**

1. The research question is important and preliminary. Trying to evaluate LLM's potential of SFT without training sounds interesting.
2. Potential-Rank is training free. It originates from the analysis of LLM's embedding eigenvalues, which is reasonable in design and theory.
3. Comprehensive study on 18 LLMs and 8 datasets testify to the effectiveness and efficiency of their method.

**Weaknesses:**

1. While the authors discuss the functionality of LCS, AGS and b_pre in section 3, the coefficients demonstrated in Table 1 have some outliers that disobey our expectations on these variables. I would appreciate it if the authors could provide a further analysis or some insight into this phenomenon.

**Questions:**

1. Does the vertical blue line signify the division of segment, i.e. AGC and LCS are computed with the values after this line?
2. While it might sound unrelated, I am curious about the influence of data contamination on Potential Rank. Would it blur the estimation?

---

> ### Author Response · Authors · 2025-11-19
>
> Dear Reviewer 6hgC
>
> Thank you for the thoughtful questions and careful reading of our manuscript. Below we address each point with additional analysis and clarification.
>
> ## Weaknesses
>
> > Q. While the authors discuss the functionality of LCS, AGS and $ b_{pre} $ in section 3, the coefficients demonstrated in Table 1 have some outliers that disobey our expectations on these variables. I would appreciate it if the authors could provide a further analysis or some insight into this phenomenon.
>
> **A**.
> Across most datasets, the coefficient signs follow the expected pattern described in Section 3 ($ g \downarrow $, $ c \uparrow $, $ b_{\text{pre}} \downarrow $). The small number of sign reversals do not indicate that the predictors behave oppositely; rather, they arise in datasets where AGS or LCS carry extremely weak response.
>
>
> - In those datasets, AGS or LCS exhibit very low variance across models, meaning they cannot meaningfully explain variation in SFT improvement.
> - As a result, the corresponding correlations with improvement are near zero, and the OLS estimates show large p-values.
> - Under such weak response, OLS becomes statistically unstable, allowing small noise to flip the estimated sign of the coefficients.
>
> Thus, the sign outliers reflect a lack of informative variation in specific datasets rather than a contradiction of the model’s overall behavior. When the predictors possess sufficient variance, their coefficient signs remain consistent with the intuition presented in Section 3. As future work, increasing the number of model observations for such low-variance datasets will further stabilize the regression estimates and reduce the likelihood of sign flips.
>
>
>
> ## Questions
>
> > Q. Does the vertical blue line signify the division of segment, i.e. AGC and LCS are computed with the values after this line?
>
> **A.**
> Yes. The vertical blue line represents the midpoint between the compression minimum and the final layer, which we define as the beginning of the “second half” of the expansion range. AGS and LCS are computed exclusively using the values after this boundary. This reflects our empirical finding (consistent with prior work) that semantic differentiation and model-to-model variability become most pronounced in the latter portion of the expansion trajectory, making this region the most informative for evaluating representational capacity.
> In the revision, we will explicitly state in the main text that AGS and LCS are computed only from the layers to the right of this boundary to avoid ambiguity.
>
>
> >Q. While it might sound unrelated, I am curious about the influence of data contamination on Potential Rank. Would it blur the estimation?
>
> **A.** We investigated this question directly and found that data contamination does not meaningfully distort Potential Rank. When we injected portions of the training data into the evaluation prompts (i.e., artificially creating contamination), the resulting AGS/LCS values and the IDS/CDS rankings were almost unchanged from the non-contaminated case.
>
> Across the datasets where we conducted this analysis (MBPP, MathQA, and ARC-Challenge), the Precision@k values and rank ordering remained effectively the same in contaminated and non-contaminated conditions. We think that our method reflects a model’s general representational adaptability—not artifacts of data leakage or memorization.
>
> We appreciate the reviewer’s careful attention to detail and the opportunity to clarify these aspects of our method. The additional analyses above have been incorporated into the revised manuscript, and we hope they help strengthen the clarity and reliability of the proposed approach.

---

### Official Review · Reviewer_J75J · 2025-11-01

**Soundness:** 3
**Presentation:** 2
**Contribution:** 3
**Rating:** 6
**Confidence:** 2

**Summary:**

The paper proposes Potential Scout, a training-free method to assess the suitability of candidate LLMs before initiating SFT. The core idea is to construct a Thinking Curve Matrix by collecting the variance trajectories of hidden representations, obtained by passing a small number of SFT samples through each layer. From this matrix, the method extracts an Activation Growth Score (AGS), representing semantic expansion capability in later layers, and a Layer Coverage Score (LCS), measuring processing stability across samples. These scores, along with pre-training performance, are used to predict post-SFT performance. The method operates as an In-dataset Scout (IDS) if prior SFT experience on the same dataset is available, or as a Cross-dataset Scout (CDS) for entirely new datasets. The variance score is calculated as the eigenvalue ratio of the Gram matrix, and models of different depths are compared fairly by interpolating their curves. The paper demonstrates that these metrics can be generated using only about 5% of the total data samples and a single forward pass, identifying top-performing models with approximately 70% accuracy across 18 LLMs and 8 benchmarks, with an analysis time of only several minutes per model.

**Strengths:**

1. The proposed method can filter candidates using a small sample set and a single forward pass significantly reduces exploration costs.

2. The paper quantifies the typical layer-by-layer pattern of compression followed by expansion, measuring these aspects separately via AGS and LCS.

3. The method has Dual-mode operation covering new domains. It is applicable as both IDS and CDS, depending on the availability of prior SFT experience.

4. The motivation for this work is well-explained, i.e., Figure 1 empirically validates the motivation for the proposed method, and its effectiveness is extensively verified across 18 LLMs and 8 datasets.

**Weaknesses:**

1. The dispersion metric (from token-by-token Gram spectra) can shift with sequence length, padding, or prompt templates, etc. I suspect that makes cross-dataset comparisons fragile unless these factors are tightly controlled.
2. Interpolating TC to compare models of different depths assumes layers are functionally aligned. In practice, later layers across architectures need functional alignment.

**Questions:**

1. Could you elaborate on the statement in Section 3.2 that "Pre-trained models initially have uniform and weekly differentiated representations, but fine-tuning induces stronger separation and specialization in these higher layers"? Specifically, what is the hypothesized or interpreted mechanism driving this representational shift during fine-tuning?
2. How robust is the proposed method to the selection of samples?
3. How does the method handle long samples?

---

> ### Author Response · Authors · 2025-11-19
>
> Dear Reviewer J75J
>
> We appreciate the reviewer’s careful reading of our manuscript and the opportunity to address these detailed and constructive comments. Below we clarify the design choices of our method, the underlying mechanisms motivating our indicators, and the additional analyses we performed in response to the reviewer’s concerns.
>
>
> ## Weaknesses
>
> > Q.
> > The dispersion metric (from token-by-token Gram spectra) can shift with sequence length, padding, or prompt templates, etc. I suspect that makes cross-dataset comparisons fragile unless these factors are tightly controlled.
>
> **A.**
> We agree that dispersion can, in principle, vary with input formatting. To minimize such effects, our implementation incorporates several safeguards:
>
> - **Seq-length normalization is built into Eq.(1).**
>   The denominator $1 - \tfrac{1}{T}$ adjusts the concentration measure so that shorter and longer sequences are normalized to the same scale.
>
> - **Prompt formats are controlled.**
>   We use each model’s official template when available; otherwise, we enforce a shared template across models.
>
> - **Padding tokens never enter the hidden states.**
>   With `padding=False` and batch size 1, pad tokens are excluded entirely from \(H\) and thus cannot affect the Gram spectrum.
>
> - **Cross-dataset stability is empirically verified.**
>   CDS operates over eight datasets with large variation in input length and structure, yet precision@k remains stable across all settings.
>
> These design choices together mitigate artifacts introduced by sequence length or prompting style.
>
>
>
> > Q.
> > Interpolating TC to compare models of different depths assumes layers are functionally aligned. In practice, later layers across architectures need functional alignment.
>
>
> **A.**
> We agree that layers across different architectures are not functionally aligned. However, the goal of our method is not to compare layer semantics but to quickly identify promising fine-tuning candidates. For this purpose, we use interpolation only as a simple depth-normalization step so that models with different numbers of layers can be compared on the same scale. Our indicators (AGS and LCS) depend solely on the shape of the overall compression–expansion trajectory—not on the function of any specific layer.
>
> At the same time, as shown in Figure 3, different architectures follow similar compression–expansion trajectories. This suggests that our simple interpolation step already captures some cross-model functional alignment, and we appreciate the suggestion to further extend our scope by exploring more explicit functional alignment strategies in future work. Such refinement would complement our lightweight scouting setting, where rapid and fair comparison remains the priority.

---

> ### Author Response · Authors · 2025-11-19
>
> ## Questions
>
> > Q.
> > Could you elaborate on the statement in Section 3.2 that "Pre-trained models initially have uniform and weakly differentiated representations, but fine-tuning induces stronger separation and specialization in these higher layers"? Specifically, what is the hypothesized or interpreted mechanism driving this representational shift during fine-tuning?
>
> **A.**
> Pre-training builds broad, general-purpose representations, whereas fine-tuning makes the model develop task-specific features—changes that naturally emerge in the higher layers. This process can be summarized as follows:
>
> - **Pre-training creates broadly useful higher-layer features.**
>   Because the model must support many domains and input types, it avoids over-specializing, leading to relatively uniform and weakly separated representations in higher layers.
>
> - **SFT specializes these layers for the target task.**
>   During SFT, the model improves its ability to separate and structure the semantic relationships emphasized in the training data, leading to sharper task-relevant distinctions in the upper layers.
>
> - **Resulting geometric effects.**
>   SFT increases dispersion when distinctions are initially weak, or reorganizes an already dispersed space into clearer task-aligned structure. In both cases, the representation manifold becomes better shaped around the supervised data.
>
>
> - **Connection to AGS/LCS.**
>   Because AGS and LCS measure slope and alignment through eigenvalue spectra, they directly capture these specialization shifts, making them effective indicators of supervised fine-tuning adaptability.
>
>
> - **Evidence from prior work.**
>   - [1] Mor Geva, Roei Schuster, Jonathan Berant, *et al.* *Transformer Feed-Forward Layers Are Key-Value Memories.* EMNLP 2021.
>   - [2] Zheng Zhao, Yftah Ziser, Shay B. Cohen, *et al.* *Layer by Layer: Uncovering Where Multi-Task Learning Happens in Instruction-Tuned Large Language Models.* EMNLP 2024.
>   - [3] Mirian Hipolito Garcia, Camille Couturier, Daniel Madrigal Diaz, *et al.* *Exploring How LLMs Capture and Represent Domain-Specific Knowledge.* arXiv:2504.16871, 2025.
>
> > Q.
> > How robust is the proposed method to the selection of samples?
>
> **A.**
> To examine this, we compared random sampling with a structured sampling strategy:
>
> - Encode all queries using Sentence-BERT
> - Cluster using HDBSCAN (chosen for not requiring the number of clusters and for stability under variable density)
> - Select one representative query per cluster
>
> Repeating IDS and CDS with these representative samples yielded performance broadly comparable to random sampling, although the random baseline occasionally suffered slight drops. In contrast, CDS exhibited modest but consistent gains in several cases (e.g., GSM8K at $k{=}3$: $0.00 \to 0.33$; OpenBookQA at $k{=}3$: $0.33 \to 0.67$), and the variance across runs decreased.
>
> This indicates that the semantic-similarity–based approach is both reasonable in design and stable in practice.
>
>
>
> > Q.
> > How does the method handle long samples?
>
> **A.**
> TCM construction removes sequence-length dependence by reducing each layer’s hidden state $H^{(\ell)} \in \mathbb{R}^{T \times D}$ to a single dispersion scalar via the eigenvalue spectrum of
> $G = HH^\top$. The normalization term $1 - \tfrac{1}{T}$ ensures comparability across long and short inputs.
>
> Additionally, several of our datasets (CoQA, LeetCode, Hendrycks Math) contain naturally long queries, yet their prediction accuracy remains on par with shorter-input tasks. This empirically confirms that the method handles long inputs effectively. We hope these clarifications address the reviewer’s concerns and help convey the rationale and robustness of our design choices.
>
> We thank the reviewer for the insightful comments, which meaningfully broadened our perspective on the design and evaluation of the method.

---

### Official Review · Reviewer_tXWb · 2025-11-01

**Soundness:** 2
**Presentation:** 2
**Contribution:** 2
**Rating:** 2
**Confidence:** 4

**Summary:**

This paper introduces Potential Scout, a framework for predicting which LLMs will benefit most from supervised fine-tuning (SFT) on a specific dataset without performing actual training. The method constructs a Thinking Curve Matrix (TCM) by tracking hidden state representations across transformer layers, then derives two diagnostic indicators: Activation Growth Score (AGS), measuring semantic expansion capability, and Layer Coverage Score (LCS), quantifying processing consistency. Combined with pre-SFT benchmark scores, these features train two regression models: In-dataset Scout (IDS) using ordinary least squares for datasets with prior SFT experience, and Cross-dataset Scout (CDS) using linear mixed models for new datasets. Evaluation across 18 LLMs and 8 datasets shows IDS achieves 70% Top-7 precision and CDS achieves 70% Top-9 precision, requiring only 5% dataset samples and 7-8 minutes per model.

**Strengths:**

**Important problem**: The observation that SFT can unpredictably harm models (Figure 1) addresses a genuine challenge with significant practical depth. The problem is real and consequential for LLM deployment.

**Comprehensive evaluation**: Testing 18 models across 8 datasets spanning math, coding, and general QA provides broad empirical coverage. The dual-scale evaluation (1-1.5B and 7-8B models) strengthens the empirical findings.

**Weaknesses:**

**Missing theoretical understanding of the problem**: While the paper observes that SFT sometimes hurts performance, it never investigates why. A fundamental reason is catastrophic forgetting—many modern LLMs undergo extensive post-training including RLHF, DPO, or other RL-based alignment. Any major change to model weights through SFT can disrupt this carefully calibrated alignment, causing performance degradation that has nothing to do with the model's "semantic expansion capability." The paper treats all performance changes as signals about model-data compatibility when they may reflect disruption of prior RL tuning. This fundamentally undermines the approach.

**Conflating PEFT and full fine-tuning**: The paper uses examples showing full fine-tuning degradation (Figure 1), but never acknowledges that PEFT methods like LoRA and full fine-tuning have completely different failure modes. A model that degrades under full fine-tuning (due to catastrophic forgetting of RL alignment) might perform well with LoRA, which modifies only a small subspace and better preserves existing capabilities. Similarly, models might improve through on-policy supervised fine-tuning but not supervised fine-tuning. The paper's diagnostic indicators cannot distinguish these fundamentally different training scenarios, yet makes universal claims about "fine-tuning potential."

**Weak theoretical justification**: The connection between AGS/LCS and SFT potential relies on vague citations to general observations (e.g., "fine-tuned models show stronger layer-wise specialization"), but why these specific eigenvalue-based dispersion metrics should predict improvement rates is never rigorously argued. Why should the slope of semantic expansion in the second half of layers predict future training dynamics? Why would processing inconsistency (high LCS) indicate improvement potential rather than simply poor model quality? The theoretical foundation is speculative handwaving. The paper never validates whether models with low AGS actually develop stronger semantic expansion after training, or whether high LCS models actually become more consistent—these are assumed without verification.

**Overly simplistic methodology**: Using ordinary least squares regression with three features is shockingly unsophisticated for capturing complex relationships between internal representations and training outcomes. No exploration of non-linear models, decision trees, gradient boosting, neural networks, or any modern ML techniques. No interaction terms between features. The choice of focusing on "second half of expansion segment" appears completely arbitrary with no systematic ablation showing this is optimal versus other layer ranges, the full trajectory, or learned segmentation. The dispersion metric itself (Equation 1) is borrowed from a 2014 paper on entropy with no justification for why this particular formulation is appropriate for predicting SFT success.

**Questions:**

1. What is the performance of simply ranking models by pre-SFT benchmark scores? Table 1 suggests bpre alone achieves correlations of 0.516 (GSM8K), 0.648 (MathQA), 0.713 (MBPP), and 0.440 (LeetCode). How much does adding AGS and LCS improve over this trivial baseline in terms of Top-K precision?

2. How do you disentangle catastrophic forgetting of RL alignment from genuine model-data incompatibility? Can you report which models in your evaluation underwent RLHF/DPO and correlate this with performance degradation? Your indicators may simply be detecting which models are most vulnerable to disrupting their alignment.

---

> ### Author Response · Authors · 2025-11-19
>
> Dear Reviewer tXWb
>
> Thank you for the thoughtful and detailed feedback. We have carefully addressed each comment below and hope that our clarifications and additional analyses sufficiently answer your questions.
>
> ## Weaknesses
>
> > 1. Missing theoretical understanding of the problem
>
> **A.**
> We acknowledge that SFT degradation can arise from multiple mechanisms, including catastrophic forgetting of RLHF/DPO alignment. Our method does not assume a single cause; rather, it is motivated by the fact that users cannot know in advance which models are alignment-fragile or stable under SFT.
>
> In practice, users rely on pre-SFT benchmark scores, but these provide no insight into how a model will actually change during fine-tuning. Prior work shows that hidden-state geometry shifts systematically before and after SFT.
>
> Motivated by this, AGS and LCS measure how actively a model’s activation geometry responds to token patterns in the fine-tuning data. Our experiments show that this internal responsiveness predicts improvement more reliably than pre-SFT benchmarks, supporting activation-geometry indicators as practical early-warning responses.
>
>
> > 2. Conflating PEFT and full fine-tuning
>
> **A.**
> Our study focuses on SFT and full fine-tuning because these are the settings most users attempt first and because FFT exposes representational restructuring most clearly. Since FFT updates all parameters, it provides the clearest response of how a model’s internal geometry responds to new data.
>
> LoRA and other PEFT methods modify only a small subspace, reducing catastrophic shifts but largely due to limited parameter movement rather than better alignment with the dataset. For this reason, FFT remains the more informative scenario for diagnosing whether a model has room for meaningful adaptation.
>
> We do not claim that AGS/LCS generalize to all tuning strategies; throughout the paper, *fine-tuning* specifically refers to SFT. Extending these diagnostics to PEFT, RL, or on-policy tuning is an important direction for future work.
>
>
> > 3. Weak theoretical justification
>
> **A.**
> Our theoretical motivation is that the model’s activation geometry before SFT already reveals how strongly it responds to the token patterns present in the fine-tuning data. AGS and LCS summarize this responsiveness and thus indicate how much meaningful semantic structure the model has formed prior to training.
>
> - **Semantic structure emerges in deeper layers.**
>   Prior work shows that semantic differentiation mainly appears in upper Transformer layers, making the second half of the expansion trajectory the most informative region for assessing model–dataset interaction.
>
> - **Low AGS / high LCS reflect weak or inconsistent responses.**
>   Low AGS suggests insufficient semantic separation for the target data, while high LCS responses inconsistent contextual processing—both indicating that the model is not yet capturing the semantic distinctions in the training data reliably.
>
> - **Pre-SFT geometry predicts improvement.**
> Although AGS and LCS may move in either direction after SFT in models that are already strongly aligned through RL. their values still carry useful information. When combined with pre-SFT benchmark scores, these indicators provide a clearer and more reliable estimate of how much SFT will help. In practice, models with weaker pre-SFT geometry tend to gain more from SFT, showing that AGS and LCS are most effective when interpreted together with benchmark performance.
>
>
>
> > 4. Overly simplistic methodology
>
> **A.**
> Our methodology is intentionally simple because the scouting scenario requires a fast, low-cost predictor. With only 18 models per dataset, complex architectures would offer little benefit and likely overfit, whereas even a linear model with lightweight indicators already provides stable and competitive performance. Since diagnostic research in this area is still at an early stage and available data are limited, our next step is to collect more model observations and explore higher-capacity predictors as richer diagnostic tools.
>
> We focus on the second half of the expansion trajectory because this is where Transformer representations show the strongest semantic differentiation. Prior work consistently finds that meaningful activation patterns and task-relevant geometric variation emerge in deeper layers, making this region the most informative for measuring model–dataset interaction.
>
> Eigenvalue-based metrics such as AGS and LCS compactly capture dispersion, alignment, and dominant semantic directions, following a well-established line of work using spectral geometry to characterize representational capacity. Within the constraints of scouting, these simple geometric indicators offer a practical and effective design choice.

---

> ### Author Response · Authors · 2025-11-19
>
> ## Questions
>
>
> > **Q.**
> > What is the performance of simply ranking models by pre-SFT benchmark scores? Table 1 suggests \(b_{\text{pre}}\) alone achieves correlations of 0.516 (GSM8K), 0.648 (MathQA), 0.713 (MBPP), and 0.440 (LeetCode). How much does adding AGS and LCS improve over this trivial baseline in terms of Top-K precision?
>
> **A.**
> Using benchmark scores alone yields limited predictive power. For Top-7 precision, the benchmark-only baseline performs as follows:
>
> | Method           | GSM8K | MBPP | ARC-Challenge |
> |------------------|:-----:|:----:|:-------------:|
> | Benchmark-only   | 0.14  | 0.00 |     0.71      |
> | All three (IDS)  | 0.29  | 0.57 |     0.86      |
>
> The gains are substantial, particularly for MBPP where the baseline provides no useful signal. This demonstrates that AGS and LCS contribute meaningful predictive information beyond pre-SFT accuracy.
>
>
> > **Q.**
> > How do you disentangle catastrophic forgetting of RL alignment from genuine model–data incompatibility? Can you report which models in your evaluation underwent RLHF/DPO and correlate this with performance degradation? Your indicators may simply be detecting which models are most vulnerable to disrupting their alignment.
>
> **A.**
> We acknowledge that, as the reviewer points out, our indicators will partially reflect alignment fragility to some extent—RLHF/DPO-aligned models can indeed exhibit forgetting during SFT. However, users have no reliable way to know which models are alignment-sensitive, as alignment strength is generally not observable from publicly available information. In practice, the only accessible response is the pre-SFT benchmark score, which does not reveal how stable a model will be under fine-tuning.
>
> Our goal, therefore, is not to separate alignment forgetting from model–data mismatch, but to ask whether these unknown risks can be inferred from the model’s internal representations. AGS and LCS should be interpreted in this context: they do not measure model–dataset compatibility, but rather how actively and consistently the model responds to the token-level semantics present in the SFT dataset. Low AGS indicates insufficient semantic expansion, and high LCS indicates inconsistent semantic separation across samples. When combined with pre-SFT benchmark scores, these responses reliably predicted which models will change more substantially during SFT.
>
> Ultimately, our study demonstrates that activation geometry provides a practical way to estimate SFT instability in advance—regardless of whether the underlying cause is alignment fragility, dataset mismatch, or representational saturation.
>
> --------
> ### RLHF/DPO Model List
> Below we list which of the 18 models underwent RLHF or DPO according to public documentation:
>
> | Model                     | RLHF/DPO Applied? |
> |---------------------------|-------------------|
> | Qwen2.5-1.5B-Instruct     | Yes               |
> | Qwen2.5-Math-1.5B         | Yes               |
> | Qwen2.5-Coder-1.5B        | Yes               |
> | Llama-3.2-1B-Instruct     | Yes               |
> | Falcon3-1B-Instruct       | Unspecified       |
> | OLMo-SFT                  | No                |
> | OLMo-2-1B-Instruct        | Yes               |
> | gemma3-1B-it              | Yes               |
> | deepseek-coder-1.3b       | Unspecified       |
> | Qwen2.5-7B-Instruct       | Yes               |
> | Qwen2.5-Math-7B           | Yes               |
> | Qwen2.5-Coder-7B          | Yes               |
> | Llama-3.1-8B-Instruct     | Yes               |
> | Falcon3-7B-Instruct       | Unspecified       |
> | deepseek-llm-7b           | No                |
> | deepseek-math-7b          | Yes               |
> | deepseek-coder-6.7b       | Unspecified       |
> | OLMo2-7B-Instruct         | Yes               |
>
> We hope this clarifies that our indicators do not aim to isolate specific causes of degradation, but instead provide a unified and practical estimate of SFT stability that users can access before performing any fine-tuning. We thank the reviewer again for the insightful questions and constructive feedback, which helped us strengthen the motivation and interpretation of our method.

---

> > ### Comment · Reviewer_tXWb · 2025-11-27
> >
> > Thanks to the authors the detailed and helpful response.
> >
> > I have a better understanding of the intuition behind AGS and LCS metric, and do agree that it has value specifically for full supervised finetuning scenario.
> >
> >
> > Though, my concerns regarding coupling full SFT with PEFT finetuning remains. That degradation in full SFT may not indicate similar degradation in PEFT training. And the same concern applies for reinforcement finetuning, where according to latest literature, has much smaller weight updates and will suffer less catastrophic forgetting.
> >
> >
> > -------
> > I have raised rating in appreciation of author engagement, and the value I see in the method specifically for full SFT setting. My recommendation for authors is to take on the problem, and connect your solution from the perspective to catastrophic forgetting and continual learning; and clearly focus the scope of the method to SFT training.

---

> > > ### Author Response · Authors · 2025-11-28
> > >
> > > Thank you again for your helpful comments and for raising the rating. We really appreciate your thoughtful engagement with our work.
> > >
> > > We agree that SFT works differently from PEFT and RL, especially in how much the model changes and how forgetting happens. Because of these differences, the degradation we see in full SFT may not appear in the same way in those settings. We plan to look more closely at PEFT and RL and explore how our method could be adapted or extended to work there as well.
> > >
> > > If you have any other questions or thoughts at any time, please feel free to reach out. We would be happy to continue the conversation.
> > >
> > > Thank you again for your valuable feedback

---

### Official Review · Reviewer_m8FE · 2025-11-02

**Soundness:** 3
**Presentation:** 2
**Contribution:** 2
**Rating:** 2
**Confidence:** 4

**Summary:**

This paper proposes Potential Scout, a framework for assessing the fine-tuning potential of large language models (LLMs) without performing full supervised fine-tuning (SFT). The method introduces two diagnostic metrics derived from hidden-state dynamics: Activation Growth Score (AGS) and Layer Coverage Score (LCS), which capture semantic expansion and representational consistency across layers. Experiments involve 18 open-source LLMs and 8 datasets. The study shows that AGS and LCS correlate with post-SFT improvements, and that the learned Scout models can predict which LLMs will benefit most from SFT.

**Strengths:**

1. The main research question in this paper is very important in this field.
2. The designs of Activation Growth Score (AGS) and Layer Coverage Score (LCS) are novel and reasonable.

**Weaknesses:**

1. **The contribution is limited**. The proposed methods (IDS and CDS) require SFT results from multiple models on the same dataset, which means we still need to perform SFT first. This conflicts with the motivation of this paper. Even with 18 LLMs, IDS only reaches about 70% Top-7 precision (Fig. 5), suggesting that the cost saving is limited and mainly applicable to organizations that can afford many exploratory SFT runs. Small teams or new domains may still be unable to apply this method.
2. **The methodology is not clearly written**:
    1. In Eq. (2) the selection rule for $k$ and $l$ ("second half of the expansion range”) is underspecified, which makes Section 3.2 hard to follow.
    2. Section 3.4 introduces $\Delta_{i,j}$, but does not immediately describe the full pipeline from predicted $\Delta_{i,j}$ to model ranking, nor the minimal number of models required to train IDS/CDS. These details only become partially mentioned in Section 4.3–4.4. The clarity needs to be improved.
    3. According to the context in Section 3.4, CDS uses cross-dataset generalization; however, such details can not be derived from Eq. (5).

**Questions:**

1. What is the minimal number of models required to obtain a stable IDS?
2. What is the performance of a random Top-k baseline?

---

> ### Author Response · Authors · 2025-11-19
>
> Dear Reviewer m8FE
>
> Thank you for your thoughtful and constructive review. We appreciate the opportunity to clarify the motivation, methodology, and scope of our contributions.
>
> ## Weakness
> >1. Contribution Concerns
>
> **A.**
>
> We acknowledge that the In-dataset Scout (IDS) requires pre-/post-SFT benchmark scores from multiple models on the same dataset. This naturally makes IDS more suitable for organizations that already perform several exploratory SFT runs. However, this limitation is precisely why we propose the Cross-dataset Scout (CDS):
>
> - CDS does **not** require any SFT runs on the target dataset.
> - The SFT-before/after benchmark scores used in CDS come from other datasets, and these values are readily available in model cards, papers, and leaderboards, thus user can minimize additional SFT cost.
> - CDS only requires extracting AGS/LCS from hidden states on the target dataset; this takes only a few minutes per model.
> - As a result, CDS is also applicable to small teams, new domains, and cold-start settings where no prior SFT runs exist.
>
> > 2. Methodology Clarity
>
> > Q2.1. In Eq. (2) the selection rule for and ("second half of the expansion range”) is underspecified, which makes Section 3.2 hard to follow.
>
> **A.**
>
> We revised Eq. (2) to explicitly specify how the starting layer \(k\) is selected within the expansion range. In the original version, the text mentioned using the “second half of the expansion range,” but the equation did not clarify how \(k\) was chosen. To remove this ambiguity, we revised the equation by adding an explicit definition:
>
> $
> k = \text{middle layer of the expansion segment}.
> $
>
> The revised equation now appears as:
>
> \begin{equation}
> \text{AGS} = \frac{1}{N} \sum_{i=1}^{N}
> \frac{x_i^{(\ell)} - x_i^{(k)}}{\ell - k},
> \qquad
> k = \text{middle layer of the expansion segment}.
> \end{equation}
>
> This addition makes clear that the expansion range begins at the midpoint layer, aligning the mathematical formulation with the textual description in Section 3.2.
>
>
> >Q2.2. Section 3.4 introduces the method, but it does not clearly explain the full pipeline—from using the predicted values to producing the final model ranking—nor does it specify the minimal number of models required to train IDS/CDS. These details only appear partially in Sections 4.3–4.4.
>
> **A.**
>
> We have revised Section 3.4 to more clearly describe the overall prediction pipeline.
> Specifically, we now detail how AGS/LCS and pre-SFT scores are extracted, how IDS and CDS models
> predict the expected improvement, and how these predictions are translated into estimated post-SFT
> performance to generate the final model ranking. In addition, we added a new paragraph titled
> **Overall Prediction Pipeline** and explicitly included the sentence:
> ``IDS therefore learns dataset-specific regressions that relate our indicators to fine-tuning gains
> and becomes stable once roughly ten model observations are available,''
> which clarifies the point that IDS stabilizes after approximately ten model observations—a fact that
> was previously mentioned only indirectly in Sections 4.3-4.4.
>
>
> > Q2.3. According to the context in Section 3.4, CDS uses cross-dataset generalization; however, such details cannot be derived from Eq. (5).
>
> **A.**
>
> To make the generalization mechanism of CDS more explicit, we updated both the equation and the text. In particular, we added the condition
>
> $
> \gamma_{\text{unseen}} = 0
> $
>
> directly below Eq. (5). This modification clarifies that, for datasets not observed during training, the CDS predictions are based solely on the fixed-effect components.
> We also revised the surrounding paragraph in Section 3.4 so that the textual explanation explicitly matches this formulation, ensuring that the description of cross-dataset generalization is fully aligned with the updated equation.

---

> ### Author Response · Authors · 2025-11-19
>
> ## Questions
>
>
> > Q. What is the minimal number of models required to obtain a stable IDS?
>
> **A.**
>
> To determine the stability requirements, we conducted the following experiment:
>
> - Fix 5 models as a held-out evaluation set.
> - From the remaining 13 models, randomly sample training sets of size \{3, 5, 7, 9, 11\}.
> - For each size, repeat 100 times, train IDS, and compute Precision@3 on the unseen models.
>
>
> Across all 8 datasets (GSM8K, MATH, MBPP, LeetCode, OpenBookQA, ARC-Challenge, CoQA, MathQA),
> the following pattern emerges:
>
> Key observations from the figure:
>
> | **Dataset**      | **3 models** | **5 models** | **7 models** | **9 models** | **11 models** |
> |------------------|--------------|--------------|--------------|--------------|---------------|
> | GSM8K            | 0.27         | 0.26         | 0.26         | 0.25         | 0.21          |
> | MATH             | 0.21         | 0.21         | 0.20         | 0.16         | 0.14          |
> | MBPP             | 0.07         | 0.09         | 0.09         | 0.03         | 0.00          |
> | LeetCode         | 0.31         | 0.31         | 0.32         | 0.31         | 0.22          |
> | OpenBookQA       | 0.13         | 0.17         | 0.13         | 0.08         | 0.06          |
> | ARC-Challenge    | 0.18         | 0.17         | 0.11         | 0.10         | 0.00          |
> | CoQA             | 0.26         | 0.18         | 0.16         | 0.12         | 0.13          |
> | MathQA           | 0.20         | 0.17         | 0.10         | 0.10         | 0.09          |
>
>
> **Using 9-11 training models is sufficient for stable IDS predictions.**
> Across datasets, variance is very high when using 3-5 models and remains unstable at 7 models.
> By contrast, most datasets enter a **practically stable region** (typically 0.20 or below)
> once **9-11 models** are used, and adding more models often yields only marginal improvement.
>
>
>
>
>
> > Q. What is the performance of a random Top-k baseline?
>
> **A.**
>
> A random baseline yields low precision:
> - Precision@7 ≈ 0.30, Precision@9 ≈ 0.50
>
> **In contrast, our methods achieve:**
> - **IDS:** Precision@7 ≈ 0.70
> - **CDS:** Precision@7 ≈ 0.55
>
> which corresponds to roughly a **2× improvement** over random selection.
> Moreover, IDS and CDS consistently place the best-performing model within the top 1-3, whereas random selection rarely identifies the best model unless \(k\) is very large.
>
> We sincerely thank the reviewer for the valuable feedback and insightful suggestions, which have significantly improved the clarity and quality of our manuscript.

---

### Meta-Review · Area_Chair_eCYj · 2026-01-06

**Summary:**

The paper explores a method to predict the performance of an LLM after being fune-tuned on a dataset. The authors provide an estimate based on previous training jobs that allows for a cheap estimate without performing the (often costly) fine-tuning job.
The problem tackled is mentioned by all reviews to be relevant and motivated. The design is said to be sensible (e.g. m8FE  “The designs of Activation Growth Score (AGS) and Layer Coverage Score (LCS) are novel and reasonable.”). Additionally, the reviews appreciated the wide range of experiments conducted to evaluate the method (e.g. 6hgC “Comprehensive study on 18 LLMs and 8 datasets testify to the effectiveness and efficiency of their method.”).

The reviews raised a few concerns, where I found two to be major. The first is by m8FE that mentioned the limited applicability of the method .The IDS method requires extensive experiments on the same dataset, making it non-applicable to low budget teams or new domains. The authors acknowledged this (they specify it in the paper as well) and mentioned that CDS can be used in such cases. However, the performance of CDS is lesser than that of IDS. The authors do show it is better than a random baseline (70% top-9 precision vs 50% for random), but this does raise the question of whether a simple baseline (not as naive as random) can do as well. My takeaway from the discussion is not necessarily that the methods are limited but that this deserves more exploration in the paper.

The second issue relates to an over simplistic approach, or rather the lack of evidence that the choices made are correct. Reviewer tXWb mentioned ways for the 3 signals could have been enriched and other ways for their values to be used for a prediction (see paragraph “Overly simplistic methodology”). Reviewer J75J mentioned factors that could be better controlled in CDS and the thinking curve (TC) can adapt to different depths. The authors’ responses for these questions were explanations after the fact, but I believe that a thorough work should have some ablation or analysis justifying such choices.

Due to these issues I believe the paper is not yet ready for publication. I would like to note that the main content is here: The method is motivated and sensible, and the overall results are promising, so a version integrating the suggested improvements is likely to be a good paper worthy of being published in a top venue.

**Reviewer Concerns:**

I do not think the concerns were sufficiently mitigated in the rebuttal

**Reviewer Scores:**

Its unlikely the reviewers m8FE, tXWb would modify their scores to 6. Maybe to 4.
The other two are likely to remain with the score of 6.

---

### Decision · Program_Chairs · 2026-01-26

Reject